# P4Q: Learning to Prompt for Quantization in Visual-language Models

## Abstract

Large-scale pre-trained Vision-Language Models (VLMs) have gained prominence in various visual and multimodal tasks, yet the deployment of VLMs on resource-constrained platforms remains challenging due to their prohibitive computational and memory overhead. Quantization of VLMs can substantially reduce the computational and memory costs, which are in urgent need. There are two prevailing paradigms, Quantization-Aware Training (QAT) can effectively quantize large-scale VLMs but incur a huge training cost, while low-bit Post-Training Quantization (PTQ) suffers from a notable performance drop. We propose a 'Prompt for Quantization" (P4Q) method, in which we design a lightweight architecture to leverage contrastive loss supervision to enhance the recognition performance of a PTQ model. Our method can effectively reduce the gap between image features and text features caused by low-bit quantization, based on learnable prompts to reorganize textual representations and a low-bit adapter to realign the distributions of image and text features. We also introduce a distillation loss based on cosine similarity predictions to distill the quantized model using a full-precision teacher. Extensive experimental results demonstrate that our P4Q method outperforms prior arts, even achieving comparable results to its full-precision counterparts. For instance, our 8-bit P4Q can theoretically compress the CLIP-ViT/B-32 by $4 \times$ while achieving 79.42% Top-1 accuracy, outperforming the learnable prompt fine-tuned full-precision model by 2.91% with negligible additional parameters on the CIFAR100 dataset. Test code and checkpoints are available at `https://anonymous.4open.science/r/ICLR2024-P4Q-1255`.

## 1 Introduction

Recent research in large-scale pre-trained Vision-Language Models (VLMs) (Radford et al., 2021; OpenAI, 2023; Brown et al., 2020) have achieved phenomenal performance in both computer vision and natural language processing, demonstrating their potential in learning open-world concepts. However, the deployment of these models on downstream resource-constrained devices presents challenges due to their high computational and storage requirements. For example, the CLIP-ViT/B-32 (Radford et al., 2021) contains 152-MB parameters and demands 78G FLOPs for inference. Therefore, VLMs compression becomes an urgent requirement.

Substantial efforts on network compression have been made towards efficient online inference (Xu et al., 2022; Romero et al., 2014; Qin et al., 2020; Denil et al., 2013). Quantization, in particular, offers compatibility with resource-limited devices by compressing networks into low-bit formats. There are two prevailing quantization methods, Quantization-Aware Training (QAT) (Li et al., 2022; Xu et al., 2023) and Post-Training Quantization (PTQ) (Lin et al., 2021). While QAT usually outperforms PTQ, it is necessary to train and optimize all parameters of the model in the quantization process. The extensive expert knowledge and considerable GPU resources for training make QAT impractical for VLMs. Additionally, the pre-training data for VLMs are often huge and unpublished, such as 400 million samples for CLIP. In contrast, PTQ quantizes the models by directly computing quantization parameters during the inference of pre-trained full-precision models, eliminating the need for large-scale pre-training data. Therefore, PTQ is a more efficient method for quantizing VLMs.

To this standpoint, we first establish a PTQ baseline on CLIP (Radford et al., 2021) to explore the low-bit quantization of VLMs. Through an empirical study, we observe significant performance

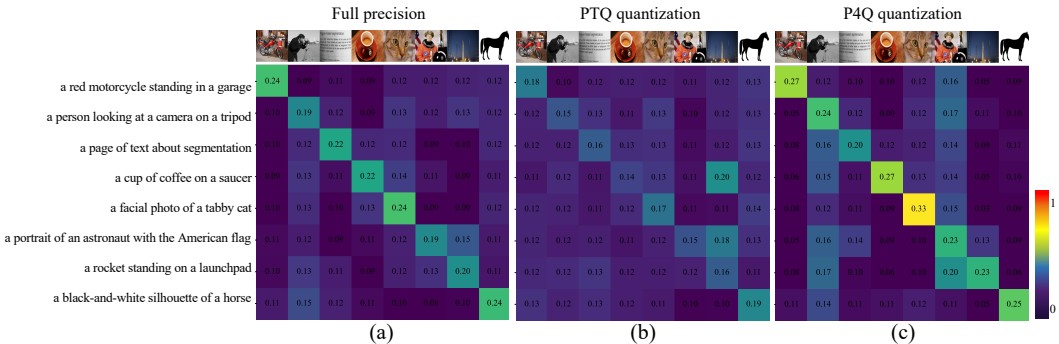

Figure 1: Cosine similarity predictions between text and image features. The horizontal axis contains 8 images, their descriptive text is vertically given in order. Each value corresponds to the similarity between the related image features and text features. The brighter the grid, the stronger the similarity between the image and the text on the current horizontal and vertical coordinates. (a) Image features and text features are encoded by full-precision CLIP. (b) Image features and text features are encoded by PTQ quantized CLIP. (c) Image features and text features are encoded by P4Q quantized CLIP.

drops on the CIFAR100 dataset as shown in Table 3a. Specifically, a 4-bit quantized CLIP-ViT/B-32 using PTQ achieves only 46.05% Top-1 accuracy, resulting in a 19.29% performance decline compared to its real-valued counterpart. We find the performance drop of quantized CLIP lies in the degraded similarity between the image features and text features. As displayed in Fig. 1 (a), diagonal grids show the brightest color, indicating that image and text features encoded by full-precision CLIP can be successfully aligned. After quantization using PTQ, the image features are partially mismatched with the descriptive text features, as shown in Fig. 1 (b). For example, the image of the rocket erroneously match the description "a cup of coffee on a saucer". Since there is no backpropagation in the quantization process of PTQ, it is similar to an open-loop system, *i.e.*, there is no effective cross-mode supervision on it. PTQ quantizes the image encoder and the text encoder separately, which causes the cross-modal gap in CLIP to be further amplified by PTQ (shown in Sec. 3).

In light of the above issue, we propose a "Prompt for Quantization" (P4Q) model, as shown in Fig. 2. P4Q introduces a light-weight architecture that leverages a joint supervision to align the image and text features of a PTQ model. Prompt tuning, which has proven effective in reorganizing textual representations in VLMs for downstream tasks (Zhou et al., 2022; Du et al., 2022), inspires our approach. P4Q trains a prompt for the quantized CLIP model, which serves to calibrate the quantized text features. To achieve this end, a low-bit and lightweight adapter (named QAdapter) is added after the image encoder, aiming to align image features with the corresponding text features. Both the prompt and the QAdapter are trained using a contrastive loss between the image features and text features. Moreover, P4Q proposes a knowledge distillation scheme to distill the quantized model using a full-precision CLIP based on the similarity predictions, so that the distribution of the low-bit model is guided by the full-precision model, which contributes to the generalization of the low-bit model. The resulting similarity predictions in P4Q are shown in Fig. 1 (c), which indicates the quantized image feature and quantized text feature are realigned by P4Q. Our contributions are summarized as follows:

1. We introduce a "Prompt for Quantization" (P4Q) method to find a suitable prompt, which can significantly improve the recognition performance of the PTQ model with a well-designed low-bit adapter (QAdapter).

2. We achieve a knowledge distillation scheme to distill the quantized model using a full-precision CLIP based on the similarity predictions. As a result, our method can effectively reduce the gap between image features and text features caused by low-bit quantization.

3. Our P4Q, for the first time, explores a promising way toward highly accurate low-bit CLIP. Extensive experiments on various benchmarks show that P4Q outperforms the PTQ models by a large margin. The 8-bit P4Q model achieves 4× compression and outperforms its learnable prompt fine-tuned full-precision counterpart on CIFAR100, validating its prospect as a general solution for the quantization of VLMs.

## 2 RELATED WORK

### 2.1 QUANTIZATION

Quantized neural networks often possess low-bit weights and activations to accelerate model inference and save memory. The commonly used model quantization methods include quantization-aware training (QAT), and post-training quantization (PTQ). In QAT, Zhang et al. (2021) builds a binarized convolutional neural network based on a projection function and a new updated rule during the backpropagation. Li et al. (2022) proposed an information rectification module and distribution-guided distillation to push the bit-width in a quantized vision transformer. While QAT often requires high-level expert knowledge and huge GPU resources for training or fine-tuning, especially the large-scale pre-trained model. To reduce the above costs of quantization, PTQ, which is training-free, has received more widespread attention and lots of excellent works arise. MinMax, EMA (Jacob et al., 2018), Percentile (Li et al., 2019), and OMSE (Choukroun et al., 2019) methods are commonly used to compress or reduce the weights of the PTQ model. MinMax normalizes the weights and bias values in the model to a predefined range, such as [-1, 1], to reduce the storage space and increase the inference speed. EMA (Jacob et al., 2018) uses a sliding average to recompute the values of some parameters in the model to a new value. By removing some low-weighted neurons, the size and computational cost of the model can be reduced in Percentile (Li et al., 2019), which can be implemented based on a threshold that depends on the percentage of weights. OMSE (Choukroun et al., 2019) quantizes the model parameters using pruning methods to reduce the computational complexity and size of the model. FQ-ViT (Lin et al., 2022) propose Log-Int-Softmax to sustain that and simplify inference by using 4-bit quantization and the BitShift operator. But the traditional PTQ will cause great performance degradation. This paper proposes a prompt tuning method for PTQ.

### 2.2 PROMPT TUNING

Prompt tuning originates from the NLP. The high-level idea of prompting is to apply a function to modify the input text, so that the language model gets additional information about the task. Concretely, given a pre-trained language model, the task is often formulated as a "fill-in-the-blank" cloze test, such as asking the model to predict the masked token in "No reason to watch. It was [MASK]" as either "positive" or "negative" for sentiment classification. The key lies in how to design the underlined part. However, the design of a prompting function is challenging and requires heuristics. Thus, continuous prompt learning methods (Zhong et al., 2021; Li & Liang, 2021), which seek to address this problem by applying trainable prompts in a continuous space. A drawback of such methods compared to searching discrete tokens is the lack of a clear way to visualize what "words" are learned for the vectors. Recent works (Zhou et al., 2022; Wang et al., 2023) adopt prompt tuning to Vision-Language Models (VLMs), such as CoOp (Zhou et al., 2022). CLIP (Radford et al., 2021) is a kind of VLM, which is trained under a large number of images and their text descriptions. It adopts a two-stream architecture, consisting of an image encoder and a text encoder that encode image and text inputs separately and produce individual vision and language representations embedded in a joint space using a contrastive loss. CoOp adopts prompt tuning to CLIP while freezing the encoders. Besides, prompt tuning has been applied to many downstream tasks of VLMs such as object detection (Du et al., 2022; Gu et al., 2021), semantic segmentation (Xu et al., 2021), and so on. Inspired by the success of prompt tuning in other tasks, this paper attempts to calibrate the distribution of the activations in the quantized VLMs by prompt. The experiment proves that it can improve the performance of quantized VLMs effectively and efficiently.

## 3 METHODOLOGY

In this section, we first revisit PTQ and CLIP in Sec. 3.1. Then, we describe our "Prompt for Quantization" (P4Q) method in Sec. 3.2. The overview of our framework is given in Fig. 2. In Sec. 3.3, the optimization process is shown.

### 3.1 PRELIMINARIES

**Post Training Quantization (PTQ).** Assuming the quantization bit-width is $b$, the quantizer $Q(\mathbf{x}|b)$ can be formulated as a function that maps a floating-point number $\mathbf{x} \in \mathbb{R}$ to the nearest quantization bin:

$$Q(\mathbf{x}|b) : \mathbb{R} \to \hat{\mathbf{x}}, \tag{1}$$

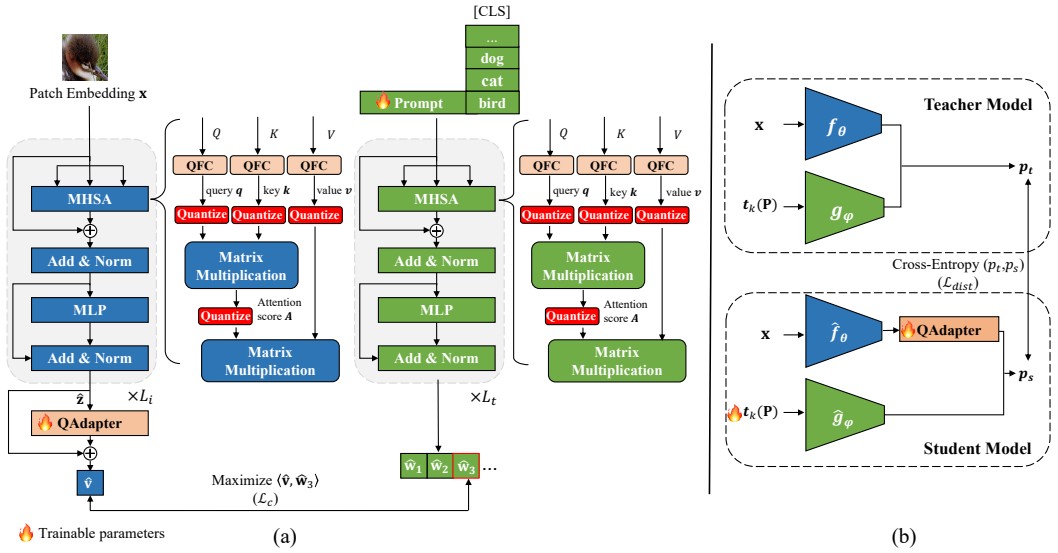

Figure 2: Overview of P4Q. The blue parts represent the visual stream, and the green parts represent the textual stream. The learnable parameters are marked by the icon of fire. QFC denotes the quantized fully connected layer. Figure (a) shows the structure of quantized CLIP. Figure (b) shows the knowledge distillation module. $f_\theta$ and $g_\psi$ represent the full-precision image and text encoders respectively. $\hat{f}_\theta$ and $\hat{g}_\psi$ represent the low-bit image and text encoders respectively.

$$\hat{\mathbf{x}} = \begin{cases} \{-2^{b-1}, \cdots, 2^{b-1} - 1\} & \text{Signed,} \\ \{0 \cdots, 2^b - 1\} & \text{Unsigned.} \end{cases} \tag{2}$$

There are various quantizer $Q(\mathbf{x}|b)$, where uniform (Jacob et al., 2018) are typically used. Uniform quantization is well supported on most hardware platforms. Its unsigned quantizer $Q(\mathbf{x}|b)$ can be defined as:

$$Q(\mathbf{x}|b) = \text{clip}(\lfloor \frac{\mathbf{x}}{s_\mathbf{x}} \rceil + zp_\mathbf{x}, 0, 2^b - 1), \tag{3}$$

where $s_\mathbf{x}$ (scale) and $zp_\mathbf{x}$ (zero-point) are quantization parameters.

In the experiments, we use the OMSE (Choukroun et al., 2019) method to get $s_\mathbf{x}$ and $zp_\mathbf{x}$:

$$z_\mathbf{x} = \frac{1}{n} \sum_{i=1}^n (x_i - \bar{x})^2, \Delta_\mathbf{x} = \sqrt{z_\mathbf{x}}, \tag{4}$$

$$s_\mathbf{x} = \frac{2\Delta_\mathbf{x}}{2^b - 1}, \ zp_\mathbf{x} = \lfloor \frac{\text{round}(z_\mathbf{x})}{2} \rfloor, \tag{5}$$

where $z_\mathbf{x}$ is the sample variance of the activations in $\mathbf{x}$, $\Delta_\mathbf{x}$ is the square root of $z_\mathbf{x}$ and n is the number of elements in $\mathbf{x}$. The calibration data, a tiny amount of data taken from the training dataset, is used to generate $s_\mathbf{x}$ and $zp_\mathbf{x}$.

The dequantization process can be formulated as:

$$\tilde{\mathbf{x}} = (\hat{\mathbf{x}} - zp_\mathbf{x}) \times s_\mathbf{x}, \tag{6}$$

**CLIP** (Radford et al., 2021). CLIP consists of an image encoder $f_\theta(\cdot)$ and a text encoder $g_\psi(\cdot)$. Specifically, the image $\mathbf{x} \in \mathbb{R}^{H \times W \times C}$ and the text $\mathbf{t} \in \mathbb{R}^D$ are fed into $f_\theta(\cdot)$ and $g_\psi(\cdot)$ respectively to obtain the image feature $\mathbf{z} \in \mathbb{R}^D$ and the text feature $\mathbf{w} \in \mathbb{R}^D$, where $\mathbf{t}$ is the input word token. In CLIP, $\mathbf{t}$ is obtained via one of the hand-crafted prompts which have a template like "a photo of a [CLS]", where [CLS] is the class name of the testing image. Thus, the probability of predicting the testing image $\mathbf{x}$ as the class $y_i$ can be computed as:

$$p(y_i|\mathbf{x}) = \frac{e^{\langle \mathbf{z}, \mathbf{w}_{y_i} \rangle / \tau}}{\sum_{k=1}^K e^{\langle \mathbf{z}, \mathbf{w}_k \rangle / \tau}}, \tag{7}$$

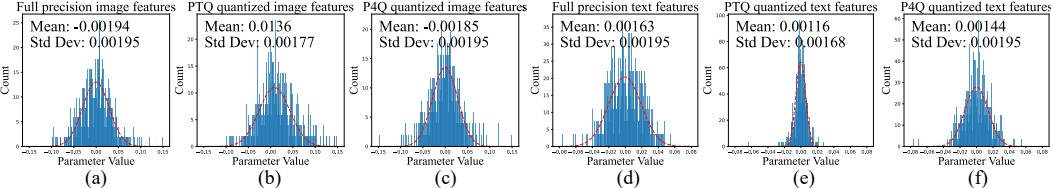

Figure 3: Histogram of image features and text features of the same class ('wolf') on CIFAR100, where the horizontal axis denotes the value of each element of the feature vectors, and the vertical axis denotes the number of elements. Mean and Ste Dev denote the mean and standard deviation of the distribution, respectively.

where $\tau$ is a temperature parameter learned by CLIP, $\langle \cdot, \cdot \rangle$ denotes the Cosine similarity, $\mathbf{w}_k$ is the feature derived from $\mathbf{t}_k$ of the $k$-th class, and $K$ is the total number of downstream dataset classes.

### 3.2 PROMPT FOR QUANTIZATION

CLIP is challenging to quantify by QAT due to its significant training burden and non-public pre-training dataset. Furthermore, naive PTQ causes significant quantization error and performance deterioration of CLIP from full-precision to low-bit. As shown in Fig. 3 (a) and (b), the distribution of image features is disturbed by PTQ. Fig. 3 (d) and (e) indicate the distortion of text features is even more serious. To alleviate the quantization error, we must calibrate the output features of the low-bit model. Since prompt tuning is effective and efficient, we suggest Prompt for Quantization (P4Q) to calibrate the output features of the low-bit CLIP quantized by PTQ. The P4Q details are introduced following.

Following CLIP, we choose ViT-B/32 (Dosovitskiy et al., 2021) and a text transformer as the image encoder and text encoder with the fixed weights. As shown in Fig. 2 (a), the image encoder and text encoder contain $L_i$ and $L_t$ blocks respectively. The self-attention module in each block is conducted quantization following PTQ in sec. 3.1. It should be noted that there is no weights training in PTQ, *i.e.*, the pre-trained full-precision weights of the encoders are directly quantized to low-bit weights. To relieve the distribution bias caused by quantization, we replace the hand-crafted prompt templates with a set of continual trainable vectors named learnable prompt $\mathbf{P}$ in the quantized textual stream. Specifically, $\mathbf{P}$ are concatenated with the embedding of a class name, formulating the text description $\mathbf{t}_k(\mathbf{P})$ of the $k$-th class as:

$$\mathbf{t}_k(\mathbf{P}) = [\mathbf{p}]_1[\mathbf{p}]_2 \ldots [\mathbf{p}]_M[\mathbf{CLS}]_k, \tag{8}$$

where each $[\mathbf{p}]_m \in \mathbb{R}^D$, $m \in \{1, \ldots, M\}$, is a learnable token of $\mathbf{P}$, and $\mathbf{P} \in \mathbb{R}^{D \times M}$ is shared among all classes. $[\mathbf{CLS}]_k$ is the text embedding of the $k$-th class name, which can also appear at the start and middle of the prompt. The text description is quantized before inputting the self-attention module.

Then the unsigned quantized text encoder produces a text feature $\hat{\mathbf{w}}_k$, which is formulated as:

$$\hat{\mathbf{w}}_k = \text{clip}(\lfloor \frac{\hat{g}_\psi(\mathbf{t}_k(\mathbf{P}))}{s_\mathbf{w}} \rceil + zp_\mathbf{w}, 0, 2^b - 1), \tag{9}$$

where $\hat{g}_\psi$ represents the quantized text encoder. $s_\mathbf{w}$ and $zp_\mathbf{w}$ are determined by full-precision $\mathbf{w}_k$, the calculation of which can be refer to sec. 3.1.

CLIP employs a two-stream structure, requiring the output feature of the image encoder to align with the output feature of the text encoder. Since the weights of the image encoder are fixed, the feature of the same image remains constant. It is difficult to change the distribution of image feature even if prompt is learnable.

Therefore, learnable parameters are in need in the image stream to jointly calibrate the image feature and the text feature. As shown in Fig. 2 (a), we attach a low-bit module, namely QAdapter, in the image stream. The QAdapter can project the low-bit image feature $\hat{\mathbf{z}}$ to the appropriate distribution based on the text feature $\hat{\mathbf{w}}_k$. The feature $\hat{\mathbf{z}}$ is formulated as:

$$\hat{\mathbf{z}} = \text{Q}(\hat{f}_\theta(\mathbf{x})|b) = \text{clip}(\lfloor \frac{\hat{f}_\theta(\mathbf{x})}{s_\mathbf{z}} \rceil + zp_\mathbf{z}, 0, 2^b - 1), \tag{10}$$

where $\hat{f}_\theta$ represents the image encoder containing the quantization module. $s_{\mathbf{z}}$ and $zp_{\mathbf{z}}$ are determined by full-precision $\mathbf{z}$. The details of QAdapter is shown in Fig. 4. It consists of two low-bit fully connected layers named QFC, forming a bottleneck structure with a short cut. The feature $\hat{\mathbf{v}}$ calculated by QAdapter is formulated as:

$$\hat{\mathbf{v}} = \alpha \hat{h}_2(\mathrm{Q}(\hat{h}_1(\hat{\mathbf{z}})|b)) + (1-\alpha)\hat{\mathbf{z}}, \tag{11}$$

where $\hat{h}_1$ and $\hat{h}_2$ represent the two QFC and $\alpha$ denotes the adapt ratio. To simplify the function, we omit the ReLu activation function. The similarity predictions between the image $\mathbf{x}_i$ and the text description of class $y_i$ is finally computed as:

$$p(y_i|\mathbf{x}_i) = \frac{e^{\langle \hat{\mathbf{v}}_i, \hat{\mathbf{w}}_{y_i}\rangle/\tau}}{\sum_{k=1}^{K} e^{\langle \hat{\mathbf{v}}_i, \hat{\mathbf{w}}_k\rangle/\tau}}. \tag{12}$$

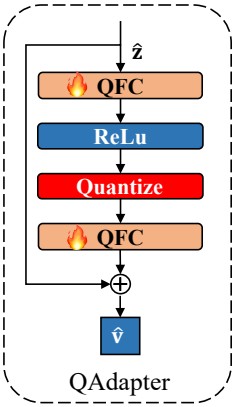

Figure 4: Overall structure of QAdapter.

### 3.3 Optimization Process of P4Q

The key to P4Q is to provide an efficient training paradigm for PTQ. It retains the high efficiency of PTQ and introduces a learnable prompt and QAdapter to improve the quantization performance. The purpose of these learnable parameters is to realign the distributions of image and text features. In this subsection, a joint loss function in supervising prompt and adapter training are detailed below.

Following the loss function of CLIP, the Cosine similarity of the quantized image feature and text feature is adopted as the classification loss $\mathcal{L}_c$. It can directly affect the distribution alignment of images and text features, which is formulated as:

$$\mathcal{L}_c = \mathbb{E}[-\log\frac{e^{\langle \hat{\mathbf{v}}_i, \hat{\mathbf{w}}_{y_i}\rangle/\tau}}{\sum_{k=1}^{K} e^{\langle \hat{\mathbf{v}}_i, \hat{\mathbf{w}}_k\rangle/\tau}}]. \tag{13}$$

To address the distributions of image and text features mismatch that occurred in the quantized CLIP, the image and text features's distributions of the full-precision model can be used as a reference. We introduce a knowledge distillation module that use similarity predictions of the full-precision teacher to optimize the learnable prompt and QAdapter. Knowledge distillation can help reduce the difficulty of optimizing the additional parameters in the quantized model and determine the optimization direction for the cross-modal alignment. The distillation loss is formulated as:

$$\mathcal{L}_{dist} = -\frac{1}{n}\sum_{i=1}^{n} p_s(y_i|\mathbf{x}_i)log(p_t(y_i|\mathbf{x}_i)) = -\frac{1}{n}\sum_{i=1}^{n} \frac{e^{\langle \hat{\mathbf{v}}_i, \hat{\mathbf{w}}_{y_i}\rangle/\tau}}{\sum_{k=1}^{K} e^{\langle \hat{\mathbf{v}}_i, \hat{\mathbf{w}}_k\rangle/\tau}} log\frac{e^{\langle \mathbf{z}_i, \mathbf{w}_{y_i}\rangle/\tau}}{\sum_{k=1}^{K} e^{\langle \mathbf{z}_i, \mathbf{w}_k\rangle/\tau}}, \tag{14}$$

where $\mathcal{L}_{dist}$ is defined as the cross entropy function between the softmax similarity predictions of full-precision encoders ($p_t$) and quantized encoders ($p_s$). The subscript $n$ denotes the batch size as shown in Fig. 2 (b).

Different from prior work (Zhuang et al., 2018) that needs to match the outputs from every intermediate block, or further using multi-step progressive structural transition (Martinez et al., 2020), we find that our distillation loss, while much simpler, can yield competitive results. The overall training losses for our approach are:

$$\mathcal{L} = \mathcal{L}_c + \lambda\mathcal{L}_{dist}, \tag{15}$$

where $\lambda$ is a trade-off hyper-parameter. The prompt and QAdapter trained by $\mathcal{L}$, calibrate the text features and image features effectively. As shown in Fig. 3 (c) and (f), the distributions of P4Q quantized image and text features are moved to their full-precision counterparts and realigned.

## 4 Experiment

In this section, we evaluate the performance of the proposed P4Q architecture. Extensive results reveal that P4Q outperforms the PTQ baseline by a considerable margin, achieving comparable results to ts full-precision counterparts.

### 4.1 Datasets and Implementation Details

**Datasets.** The experiments are carried out on the CIFAR100 (Krizhevsky et al., 2009) and ImageNet-1k (Deng et al., 2009) datasets. The CIFAR100 dataset consists of 60k images with a

size of 32×32 from 100 classes, which are split into 10 tasks with 10 classes. Each class consists of 500 training and 100 testing samples. The ImageNet-1k dataset contains samples sized 224×224 from 1000 classes. Each class consists of about 1,300 training and 50 test samples.

**Baseline and comparisons.** To the best of our knowledge, there is no publicly available source code on Post-Training Quantization (PTQ) of CLIP at this point, so we implement a PTQ baseline using OMSE (Choukroun et al., 2019). The baseline employs signed channel-wise quantization for weights and unsigned layer-wise quantization for activations following Eq. 2. We randomly sample 10 iterations of training images as the calibration data to determine the scales and zero-points in Eq. 5. We include some conventional PTQ methods as low-bit comparisons, including MinMax, EMA (Jacob et al., 2018) and Percentile (Li et al., 2019). We further introduce a supervised low-bit comparison, Quantized Linear Prob, where we quantize the parameters of the linear classifier using MinMax and other parameters using the OMSE as the PTQ baseline.

| Backbone | Method | #Bits | Top-1 | Top-5 | FLOPs(G) | Size(MB) |
|---|---|---|---|---|---|---|
| | Zero Shot CLIP | 32-32-32 | 65.34 | 89.00 | 53.188 | 577.08 |
| | + Prompt | 32-32-32 | 76.51 | 94.97 | 53.188 | 577.08 |
| | + Linear Prob | 32-32-32 | 80.50 | - | 53.188 | 577.28 |
| | MinMax | 8-8-8 | 64.70 | 88.85 | 13.297 | 146.95 |
| | EMA (Jacob et al., 2018) | 8-8-8 | 54.45 | 83.56 | 13.297 | 146.95 |
| | Percentile (Li et al., 2019) | 8-8-8 | 59.51 | 85.88 | 13.297 | 146.95 |
| | Baseline (Choukroun et al., 2019) | 8-8-8 | 64.48 | 88.88 | 13.297 | 146.95 |
| | + Quantized Linear Prob | 8-8-8 | 79.03 | - | 13.297 | 147.45 |
| | **+ P4Q** | 8-8-8 | **79.42** | **95.49** | 13.297 | 147.08 |
| | MinMax | 4-4-8 | 45.57 | 73.48 | 13.271 | 94.76 |
| | EMA (Jacob et al., 2018) | 4-4-8 | 35.65 | 66.02 | 13.271 | 94.76 |
| | Percentile (Li et al., 2019) | 4-4-8 | 43.28 | 72.20 | 13.271 | 94.76 |
| | Baseline (Choukroun et al., 2019) | 4-4-8 | 46.05 | 74.47 | 13.271 | 94.76 |
| ViT-B/32 | + Quantized Linear Prob | 4-4-8 | 66.11 | - | 13.271 | 94.78 |
| | **+ P4Q** | 4-4-8 | **69.20** | **91.26** | 13.271 | 94.89 |
| | MinMax | 3-3-8 | 16.16 | 37.310 | 13.266 | 81.96 |
| | EMA (Jacob et al., 2018) | 3-3-8 | 8.79 | 25.82 | 13.266 | 81.96 |
| | Percentile (Li et al., 2019) | 3-3-8 | 28.23 | 54.48 | 13.266 | 81.96 |
| | Baseline (Choukroun et al., 2019) | 3-3-8 | 16.79 | 38.68 | 13.266 | 81.96 |
| | + Quantized Linear Prob | 3-3-8 | 30.53 | - | 13.266 | 81.98 |
| | **+ P4Q** | 3-3-8 | **47.83** | **77.66** | 13.266 | 82.09 |
| | MinMax | 2-2-8 | 4.83 | 16.51 | 13.260 | 68.67 |
| | EMA (Jacob et al., 2018) | 2-2-8 | 2.96 | 12.58 | 13.260 | 68.67 |
| | Percentlile (Li et al., 2019) | 2-2-8 | 11.62 | 29.06 | 13.260 | 68.67 |
| | Baseline (Choukroun et al., 2019) | 2-2-8 | 5.89 | 19.48 | 13.260 | 68.67 |
| | + Quantized Linear Prob | 2-2-8 | 9.20 | - | 13.260 | 68.68 |
| | **+ P4Q** | 2-2-8 | **31.09** | **62.24** | 13.260 | 68.80 |

Table 1: Comparison of the accuracy with PTQ models on CIFAR100 dataset. #Bits (W-A-Attention) denotes the bit-width of weights, activations, and attention activations (query, key, value, and attention weights). The best results are bold.

**Experimental settings.** In the experiments, we choose ViT-B/32 (Dosovitskiy et al., 2021) and a text transformer as image and text encoder, both initialized with pre-trained full-precision CLIP weights ($L_i = L_t = 12$). In prompt, we fix $M$ at 16 and update trainable tokens using SGD optimizer (lr=5e-4, weight decay=0). In QAdapter, we use a fixed bit type of 8, default feature adapt ratio $\alpha$ at 0.2, and update QAdapter with AdamW optimizer (initial lr=1e-3). The trade-off hyper-parameter $\lambda$ of the distillation loss is set to 1. We trained P4Q for 50 epochs on each dataset. The train, validation, and calibration batch size is set to 128. We explore two adaptation methods on full-precision models: Linear Prob, which adopts a linear classifier on CLIP image features, and Prompt, which employs learnable prompt on full-precision models as detailed in Sec. 3.2, trained for 50 epochs. We implement these methods using PyTorch (Paszke et al., 2017) on 4 NVIDIA 3090Ti GPUs with 24 GB memory.

## 4.2 RESULTS ON CIFAR100

We first compare our method with low-bit comparisons on CIFAR100 (Krizhevsky et al., 2009). As shown in Table 1, the baseline suffers a severe performance drop on Top-1 accuracy (60.51%, 48.55%, 19.29%, 0.86% with 2/3/4/8-bit, respectively). Additionally, all current PTQ methods,

including MinMax, EMA (Jacob et al., 2018), Percentile (Li et al., 2019) and the baseline, fail to represent a low-bit CLIP model well.

The proposed P4Q boosts the Top-1 accuracy the of 2/3/4/8-bit baseline by 25.2%, 31.04%, 23.15%, 14.94%, and outperforms the 2/3/4/8-bit linear prob by 21.89%, 17.3%, 3.09%, 0.39%, respectively. Notably, the 8-bit P4Q achieves 79.42% Top-1 accuracy, surpassing the learnable prompt fine-tuned full-precision model by 2.91%, achieving comparable results to full-precision linear prob. Results confirm P4Q's effectiveness in achieving high accuracy while significantly compressing model size. The size compression rate of the proposed 2/3/4/8-bit P4Q model achieves 8.4×, 7×, 6.1× and 4×, respectively. Altogether, the proposed 4-bit P4Q model can theoretically accelerate the CLIP-ViT/B-32 by 4.0× on FLOPs, compressing the model size by 6.1× while achieving 69.20% Top-1 accuracy, forming a lighter model with full-precision comparable utilities.

| Backbone | Method | #Bits | Top-1 | Top-5 | FLOPs(G) | Size(MB) |
|---|---|---|---|---|---|---|
| | Zero Shot CLIP | 32-32-32 | 63.36 | 88.82 | 53.188 | 577.08 |
| | + Prompt | 32-32-32 | 64.70 | 88.85 | 53.188 | 577.08 |
| | + Linear Prob | 32-32-32 | 76.10 | - | 53.188 | 577.08 |
| | MinMax | 8-8-8 | 63.14 | 88.78 | 13.297 | 146.95 |
| | EMA (Jacob et al., 2018) | 8-8-8 | 41.51 | 71.28 | 13.297 | 146.95 |
| | Percentlile(Li et al., 2019) | 8-8-8 | 46.48 | 73.96 | 13.297 | 146.95 |
| | Baseline (Choukroun et al., 2019) | 8-8-8 | 64.48 | 88.88 | 13.297 | 146.95 |
| | + Quantized Linear Prob | 8-8-8 | 66.90 | - | 13.297 | 147.45 |
| | **+ P4Q** | 8-8-8 | **66.94** | **92.50** | 13.297 | 147.08 |
| | MinMax | 4-4-8 | 56.85 | 84.52 | 13.271 | 94.76 |
| | EMA (Jacob et al., 2018) | 4-4-8 | 36.95 | 66.49 | 13.271 | 94.76 |
| | Percentlile(Li et al., 2019) | 4-4-8 | 35.42 | 60.94 | 13.271 | 94.76 |
| ViT-B/32 | Baseline (Choukroun et al., 2019) | 4-4-8 | 46.05 | 74.47 | 13.271 | 94.76 |
| | + Quantized Linear Prob | 4-4-8 | 58.80 | - | 13.271 | 94.78 |
| | **P4Q** | 4-4-8 | **61.97** | **86.22** | 13.271 | 94.89 |
| | MinMax | 3-3-8 | 44.68 | 73.43 | 13.266 | 81.96 |
| | EMA (Jacob et al., 2018) | 3-3-8 | 28.45 | 55.47 | 13.266 | 81.96 |
| | Percentlile(Li et al., 2019) | 3-3-8 | 26.69 | 50.084 | 13.266 | 81.96 |
| | Baseline (Choukroun et al., 2019) | 3-3-8 | 45.18 | 73.85 | 13.266 | 81.96 |
| | + Quantized Linear Prob | 3-3-8 | 49.40 | - | 13.266 | 81.98 |
| | **P4Q** | 3-3-8 | **53.62** | **83.17** | 13.266 | 82.09 |
| | MinMax | 2-2-8 | 13.50 | 30.44 | 13.260 | 68.67 |
| | EMA (Jacob et al., 2018) | 2-2-8 | 9.08 | 22.39 | 13.260 | 68.67 |
| | Percentlile(Li et al., 2019) | 2-2-8 | 7.49 | 18.28 | 13.260 | 68.67 |
| | Baseline (Choukroun et al., 2019) | 2-2-8 | 15.32 | 33.73 | 13.260 | 68.67 |
| | + Quantized Linear Prob | 2-2-8 | 19.40 | - | 13.260 | 68.68 |
| | **P4Q** | 2-2-8 | **25.19** | **56.12** | 13.260 | 68.80 |

Table 2: Comparison of the accuracy with PTQ models on the ImageNet dataset. #Bits (W-A-Attention) denotes the bit-width of weights, activations, and attention activations (query, key, value, and attention weights).

### 4.3 RESULTS ON IMAGENET-1K

We further show a comparison on ImageNet-1k (Deng et al., 2009) dataset in Table 2. As shown, the baseline suffers a severe performance drop on the ImageNet-1k (50.02%, 20.16%, 8.38%, 0.86% with 2/3/4/8-bit, respectively). We then compare our P4Q with the 2/3/4/8-bit comparisons. Table 2 exhibits P4Q can increase the Top-1 accuracy the of 2/3/4/8-bit baseline by 9.87%, 8.44% and 5.01%, 3.74% under the same architecture and bit-width and surpass the 2/3/4/8-bit linear prob performance by 0.04%, 3.17%, 4.22%, 5.79% respectively. The 8-bit P4Q achieves 66.94% Top-1 accuracy, surpassing the learnable prompt fine-tuned full-precision model by 2.24%. The experiment results on ImageNet-1k also demonstrate the effectiveness of the proposed P4Q model in reducing the model size and computational cost.

P4Q focuses on the enhancement of the low-bit model's performance in downstream tasks. By adopting the PTQ approach for quantizing the CLIP encoders, P4Q minimizes alterations to the pre-trained parameters. Additionally, the low-bit model is distilled with its full-precision counterpart, to maintain its generalization performance. Analysis of these aspects is provided in the supplementary material.

### 4.4 ABLATION STUDY

In the following experiments, we explore the best-performed structure of P4Q by ablating and tuning its parts. All tests are performed on CIFAR100 dataset.

**Prompt and QAdapter.** We present the quantitative results of the proposed trainable prompt, QAdapter in Table 3a. As displayed, prompt and QAdapter improve the performance when applied alone, and the two methods further boost the performance considerably when combined. For example, the prompt improves the 4-bit baseline by 17.84% and the QAdapter achieves 20.23% improvement. Combining the prompt and QAdapter, the performance improvement achieves 23.15%. In Table 3a, we exhibit that QAdapter incurs almost no additional computation or memory cost to the quantized baseline, with a negligible increase in FLOPs and 0.13MB increase in size, and prompt almost does not induce any additional memory cost.

| Method | #Bits | Top-1 | FLOPs | Size(MB) |
|---|---|---|---|---|
| Full Precision | 32-32-32 | 65.34 | - | - |
| Baseline | 4-4-8 | 46.05 | 13.271 | 94.76 |
| +Prompt | 4-4-8 | 63.89 | 13.271 | 94.76 |
| +QAdapter | 4-4-8 | 66.28 | 13.271 | 94.89 |
| **+Both (P4Q)** | 4-4-8 | **69.20** | 13.271 | 94.89 |

| Method | $\lambda$ | Epochs | #Bits | Top-1 |
|---|---|---|---|---|
| Full Precision | - | - | 32-32-32 | 65.34 |
| Baseline | - | - | 4-4-8 | 46.05 |
| +$L_{dist}$ | - | 20 | 4-4-8 | 64.50 |
| +$L_c$ | - | 20 | 4-4-8 | 66.34 |
| **+Both (P4Q)** | 0.5 | 20 | 4-4-8 | 67.80 |
| | 1 | 20 | 4-4-8 | **68.12** |
| | 1.5 | 20 | 4-4-8 | 67.76 |

(a) Evaluating the modules of P4Q.  (b) Evaluating the losses of P4Q.

**Loss analysis.** We analyze the effectiveness of the proposed $L_c$ and $L_{dist}$ in the 4-bit P4Q, trained 20 epochs. As shown in Table 3b, using the knowledge distillation loss $L_{dist}$ and classification loss $L_c$ alone provides 20.30% increase to Top-1 accuracy in 4-bit models. Results validate the effectiveness of teacher similarity predictions and the contrastive supervision to a quantized student model. The two losses can further boost the performance considerably when combined together, achieving 1.78% increase than using the $L_c$ loss alone. We tune the trade-off parameter $\lambda$ of $L_{dist}$ and obtain the optimal performance with $\lambda=1$.

**Hyper-parameters selection.** We select the hyper-parameters prompt length $M$ and adapt ratio $\alpha$ using a 4-bit P4Q trained for 20 epochs. Model performance (Top-1 accuracy) with different hyper-parameter combinations $\{M, \alpha\}$ is presented in Fig. 5. Results indicate that performance initially improves and then declines as $\alpha$ varies from 0 to 1. This demonstrates the necessity of image feature adaptation, with full adaptation ($\alpha = 1$) outperforming the vanilla base ($\alpha = 0$). Additionally, P4Q with prompt learning exhibits stronger performance than without ($M = 0$), but full prompt learning ($M = 32$) performs worse than all alternatives. Exploring the setups, we identify $\{M, \alpha\} = \{16, 0.2\}$ as the combination that boosts P4Q's performance the most, achieving 68.12% Top-1 accuracy. Based on the ablative study above, we set hyper-parameters $M$ and $\alpha$ as 16 and 0.2 for the experiments in this paper.

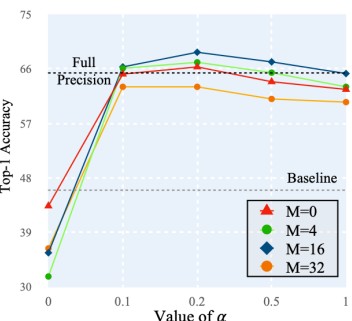

Figure 5: Effect of prompt length and adapt ratio $\{M, \alpha\}$.

## 5 CONCLUSION

This paper proposes a quantization method named "Prompt for Quantization" (P4Q) for large-scale pre-trained vision-language models. We design a low-bit and lightweight adapter with learnable prompts to significantly improve the recognition performance of a PTQ model. Besides, we propose a joint loss function including contrastive loss and distillation loss, which can effectively align the cross-modal distributions of the quantized model to their full-precision counterparts. Overall, our P4Q quantized CLIP obtains a superior performance than prior arts, even achieving comparable results to its full-precision counterparts with negligible added parameters on CIFAR100 and ImageNet-1k. In our future work, we will explore the potential of our method in other applications.

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
