# OpenReview forum: "P4Q: Learning to Prompt for Quantization in Visual-language Models"
_ICLR.cc/2024/Conference — ICLR 2024 Conference Withdrawn Submission_

### Official Review · Reviewer_xyBt · 2023-10-28

**Soundness:** 2 fair
**Presentation:** 2 fair
**Contribution:** 2 fair
**Rating:** 3
**Confidence:** 4

**Summary:**

The paper introduces a method called "Prompt for Quantization" (P4Q) to address the challenge of quantizing large-scale pre-trained Vision-Language Models (VLMs) for resource-constrained devices. It tackles the limitations of existing quantization methods, such as QAT and PTQ, which has high training costs and performance drops. P4Q leverages a lightweight architecture and a contrastive loss supervision to align image and text features in a PTQ model, effectively reducing the gap caused by low-bit quantization. Additionally, a distillation loss based on cosine similarity predictions is used to distill the quantized model using a full-precision teacher. Experimental results demonstrate the superiority of P4Q over existing methods, even achieving comparable results to full-precision models on CIFAR100.

**Strengths:**

- The paper addresses an important problem.
 - It shows that P4Q enhances the recognition performance of PTQ models, outperforming existing methods. The paper provides comprehensive experimental results on CIFAR100 and ImageNet-1k datasets, demonstrating the effectiveness of the proposed P4Q method.
 - P4Q uses a lightweight architecture, making it practical for resource-constrained platforms.

**Weaknesses:**

- Limited Novelty: The paper's approach combines established techniques, such as prompt tuning and inserting a quantized adapter, which are not particularly novel in the field of LLMs. These techniques are based on well-known and existing methods, which may limit the originality of the proposed approach. The paper incorporates existing practices like contrastive loss and distillation loss into its framework. While these methods are effective, the paper does not introduce significant innovations in terms of loss functions, potentially limiting the novelty of the proposed methodology.

- Limited Evaluation on Large Datasets: The paper's evaluation results are primarily based on the CIFAR100 dataset, which is relatively small in scale. The significant performance gap observed on the larger ImageNet-1k dataset, particularly an over 9% decrease in accuracy when compared to fp32+linear_prob, raises concerns about the effectiveness of the proposed method on larger datasets. The paper should provide further insights into what causes such a substantial performance difference.

- Given that the paper deals with the compression of multimodal models, it is essential for the authors to consider both visual and textual tasks during the evaluation phase. Currently, the paper only presents results related to image-based tasks. A more comprehensive evaluation that includes text-based tasks and cross-modal performance would provide a more holistic understanding of the proposed approach.

In light of the concerns raised, it seems that the paper may require further refinement and additional experimentation, particularly on larger datasets and cross-modal tasks, before it is ready for submission to a conference like ICLR.

**Questions:**

- What are the practical implications of implementing the P4Q method in real-world applications, and are there any potential challenges or limitations not addressed in the paper?

- How does the proposed method impact the computational and memory requirements for VLMs compared to other quantization techniques?

---

### Official Review · Reviewer_iH7g · 2023-10-29

**Soundness:** 3 good
**Presentation:** 3 good
**Contribution:** 2 fair
**Rating:** 5
**Confidence:** 4

**Summary:**

The paper proposes a method called "Prompt for Quantization" (P4Q) to address the challenges of deploying large-scale pre-trained Vision-Language Models (VLMs) on resource-constrained platforms. The method leverages contrastive loss supervision, learnable prompts, and a distillation scheme to enhance the recognition performance of low-bit quantized VLMs. Experimental results demonstrate that P4Q outperforms prior arts and achieves comparable results to full-precision models, with significant compression ratios. P4Q introduces a lightweight architecture that effectively reduces the performance gap caused by low-bit quantization by reorganizing textual representations and realigning the distributions of image and text features. The method also incorporates a distillation loss to improve the generalization of the low-bit models. Overall, P4Q shows promise in enabling the deployment of VLMs on resource-constrained platforms, offering a potential solution for various visual and multimodal tasks.

**Strengths:**

- The observation that applying an adapter directly to the output of the text encoder improves the performance of the quantized model is intriguing, as it suggests a potential domain adaptation technique to enhance results on small datasets.
- While I have some reservations about the baseline methods, the reported results in this paper appear robust.
- The simplicity and comprehensibility of the proposed method make it accessible and easy to implement.

**Weaknesses:**

# Major Concerns:
1. The proposed method appears to focus on aligning the text and image features after post-training quantization (PTQ). It is unclear whether the customized design of the learnable token $P$ and the QAdapter can, in fact, effectively improve the performance of the quantized image encoder and text encoder beyond the zero-shot classification task. Since the weights of both encoders are frozen after quantization, the proposed method seems to be closely tied to the CLIP structure and may not be generally effective for all downstream tasks. This raises concerns about the claim of "Learning to Prompt for Quantization in Visual-Language Models" and weakens the motivation of the paper.
2. The comparison with universal model quantization techniques in Table 1 and Table 2 seems unfair, as the proposed method appears to be a fine-tuning strategy specifically designed for the zero-shot classification task based on CLIP. It would be more appropriate to compare with other recent LLM-oriented model quantization methods, such as OmniQuant [1], SmoothQuant [2], Outlier Suppression+ [3], LLM-QAT [4], and QLora [5].

# Minor Issues:
1. It is recommended to use white spaces to separate the content and the caption of Figure 2 for better readability.
2. Consider using a different notation for the text feature $\hat{w}_k$, as it is commonly used to refer to weights in the context of model compression papers.

# Reference:
- [1] Omniquant: Omnidirectionally calibrated quantization for large language models. arXiv 2023
- [2] Smoothquant: Accurate and efficient post-training quantization for large language models. ICML 2023
- [3] Outlier suppression+: Accurate quantization of large language models by equivalent and optimal shifting and scaling. arXiv 2023
- [4] LLM-QAT: Data-free quantization aware training for large language models. arXiv 2023
- [5] QLora: Efficient fine-tuning of quantized LLMs. arXiv 2023

**Questions:**

- The consistency of the learnable token $P$ across different datasets is unclear, which raises concerns about its effectiveness in different domains.
- Figure 4 indicates that the input of the QAdapter remains unquantized. It would be helpful if the authors clarify whether they omitted the "Quantize" layer before the first QFC layer.
- The presence of a Gaussian-like full-precision distribution and a Laplacian-like PTQ distribution, as shown in Figure 3(e), seems unusual given the uniform quantization applied. Further explanation from the authors would be beneficial to understand this observation.
- More details on the quantization of the MHSA module, as depicted in Figure 2(a), would be valuable. It is important to know if the authors also quantized the input and output of the Softmax and how they handled the complicated activation function used in the Transformer.
- It would be worth considering the application of the classification loss $L_c$ to the PTQ methods for fine-tuning the quantization scaling factors. Direct comparison with PTQ methods may not be fair, as the proposed methods involve fine-tuning and distillation.
- I am interested in understanding whether the QAdapter and learnable $P$ improve performance on the target dataset at the expense of losing general representation capacity. It would be informative if the authors could evaluate the performance of the QAdapter and learnable $P$ trained on the CIFAR dataset when applied to the ImageNet dataset. If the results are significantly worse compared to fine-tuning on the ImageNet dataset, it may suggest overfitting of the proposed method to the target dataset, which could be contrary to the motivation of CLIP.

---

### Official Review · Reviewer_X2dx · 2023-11-03

**Soundness:** 2 fair
**Presentation:** 2 fair
**Contribution:** 1 poor
**Rating:** 3
**Confidence:** 5

**Summary:**

This paper proposes a Prompt for Quantization method that learns a prompt of CLIP to minimize the misalignment of the text embeddings and the image embeddings. They design the low-bit adapter for distribution alignment, and propose a distillation loss to minimize the difference of the text-image pair similarity between the full-precision and quantized networks.

**Strengths:**

1. The misalignment of the text feature space and the image feature space is very important in network quantization for pre-trained large vision-language models.

2. The experiments show the effectiveness of the proposed method.

**Weaknesses:**

1. The motivation is problematic. Since the CLIP model is trained in the large 4M dataset, using the images from the training set as the calibration data is harmful for the post-training quantization. Therefore, I don't think the empirical study showing the embedding space of text and images is correct because of the calibration set problem. I think the authors should also try some data-free quantization methods such as int8. Since they are free of the calibration set problem, they may get better performance on the downstream tasks.

2. The novelty is very limited. The proposed prompt engineering has be researched in a lot of former researches such as CoOp. This method seems to just use the original prompt engineering techniques in the quantized networks. What is special of the prompt engineering when being used in quantized networks?

3. The paper organization should be improved. Figure 2 does not convey sufficient information. A lot of contents such as the detailed architectures of the MSHA is redundant as they are well-know and is not related to the proposed method. I think they should be deleted.

**Questions:**

See weakness.

---

### Official Review · Reviewer_5VdB · 2023-11-07

**Soundness:** 3 good
**Presentation:** 3 good
**Contribution:** 2 fair
**Rating:** 5
**Confidence:** 4

**Summary:**

The paper studies post-training quantization (PTQ) for the pre-trained vision-language (VL) models, such as CLIP, towards fast inference speed, low overhead, and comparable accuracy to full-precision models. To bridge the gap between quantized visual/text features, the proposed method incorporates adapters and learnable prompts into vision/text encoders, respectively, and leverages a knowledge distillation framework through contrastive learning. Experimental results on three benchmark datasets were provided in terms of multiple low-bit settings.

**Strengths:**

- **Good implementation**: A sound and practical PTQ solution has been provided to compress pre-trained CLIP to low-bit models, showing promising image classification performance on several benchmark datasets.
- **Easy to follow**: The presentation of the proposed approach is straightforward and well-organized. The motivation for introducing prompts, adapters, and knowledge distillation to quantized CLIP is clearly demonstrated and supported by empirical evidence.

**Weaknesses:**

- **Less technical novelty**: Despite a concrete implementation of applying PTQ to CLIP, the technical novelties inside this paper are relatively weak -- learnable prompts, visual adapters, and quantized transformers are all well-established in prior works. Applying knowledge distillation and contrastive learning for quantization has also been explored in previous methods (e.g., *Quantization via Distillation and Contrastive Learning, TNNLS*).
- **Weak experimental design**:  The experiment is unconvincing due to the lack of 1) larger backbones (e.g., CLIP-ViT-Large, BLIP, ALBEF, etc.), 2) broad visual-language tasks (e.g., image-text retrieval, captioning, VQA, etc.), and 3) more recent PTQ methods as baselines.
- **Under-exploration of QAdapter**: The network design of the proposed QAdapter lacks insights and empirical support. It will be better to discuss and compare with some alternative implementations.
- **Unclear calibration dataset**: How to choose and design the calibration data is an essential step for PTQ approaches, especially for multimodal data. However, the paper lacks a thoughtful analysis and model discussion on the calibration data.

**Questions:**

- Can the proposed PTQ approach apply to larger pre-trained VLMs?
- How to apply the quantized CLIP given by the proposed PTQ on more general vision-language tasks? Any experimental results?